# Saturation Mutagenesis of the Transmembrane Region of HokC in *Escherichia coli* Reveals Its High Tolerance to Mutations

**DOI:** 10.3390/ijms221910359

**Published:** 2021-09-26

**Authors:** Maria Teresa Lara Ortiz, Victor Martinell García, Gabriel Del Rio

**Affiliations:** Department of Biochemistry and Structural Biology, Institute of Cellular Physiology at UNAM, Mexico City 04510, Mexico; mlara@ifc.unam.mx (M.T.L.O.); vmartinell@tutamail.com (V.M.G.)

**Keywords:** transmembrane proteins, saturation mutagenesis, deep sequencing, residue packing

## Abstract

Cells adapt to different stress conditions, such as the antibiotics presence. This adaptation sometimes is achieved by changing relevant protein positions, of which the mutability is limited by structural constrains. Understanding the basis of these constrains represent an important challenge for both basic science and potential biotechnological applications. To study these constraints, we performed a systematic saturation mutagenesis of the transmembrane region of HokC, a toxin used by Escherichia coli to control its own population, and observed that 92% of single-point mutations are tolerated and that all the non-tolerated mutations have compensatory mutations that reverse their effect. We provide experimental evidence that HokC accumulates multiple compensatory mutations that are found as correlated mutations in the HokC family multiple sequence alignment. In agreement with these observations, transmembrane proteins show higher probability to present correlated mutations and are less densely packed locally than globular proteins; previous mutagenesis results on transmembrane proteins further support our observations on the high tolerability to mutations of transmembrane regions of proteins. Thus, our experimental results reveal the HokC transmembrane region high tolerance to loss-of-function mutations that is associated with low sequence conservation and high rate of correlated mutations in the HokC family sequences alignment, which are features shared with other transmembrane proteins.

## 1. Introduction

Understanding the structure–function relationship of proteins represents a challenge to design effective pharmacological compounds [1,2]. Transmembrane (TM) proteins represent 30% of all proteins and less than 3% of these proteins have their three-dimensional atomic (3D) structures solved [3]. Most TM proteins are targets for pharmacologic intervention given their role in transport and signaling [4], thus anticipating the ability of TM proteins to adapt their sequence without affecting their activity has both basic and applied motivations. A common way to study the structure–function relationship of proteins involves the prediction of residues important for protein function based on the 3D structure of TM proteins, which are seldom available. In the absence of a 3D structure, critical residues for protein function may be predicted based on multiple sequence alignments (MSA) of similar proteins; MSA are built based on substitution matrices that, until recently, have been developed specific for TM proteins [5,6]. In either case, the precise identification of critical residues for protein function is accomplished by saturation mutagenesis of proteins, which up to date have been performed mostly on globular proteins [7,8,9,10,11,12,13,14,15,16,17,18,19,20,21,22,23,24]; a recent report on the rat neurotensin 1 D03 receptor showed that TM regions allowed for more diverse mutations than the globular regions [25].

Critical residues for protein function are commonly considered positions in a protein that upon mutation affect the folding, stability, binding, and/or catalytic activity of proteins; note that performing single-point mutations may identify loss-of-function mutations, which are mutations that eliminate protein function. We have previously reviewed the different experimental criteria used to define what a critical residue is and proposed a quantitative measurement, Criticality Index (CI), that efficiently relates protein mutations with their functional effect [26]. Several approaches have been described to predict these critical residues [27,28,29,30,31,32,33,34] and all failed to identify several known critical residues [35]. These non-predicted critical residues may be either false-positives or truly hard to predict critical residues. To filter out false-positives, especially on large-scale mutagenesis experiments of proteins, we have reported a combined experimental and computational method that CHecks for Incorrect Sequence-Phenotype Assignments, or CHISPAs [36]. ISPAs (i.e., false positives) are those protein mutants observed with both wild type and mutant phenotypes at a frequency equal or smaller than the expected experimental error introduced to generate/discover mutations. In the present study, we will use this method to study the structure–function relationship of a bitopic protein.

Bitopic proteins (i.e., having a single helical TM region) constitute a convenient model to study the structure–function relationship of TM proteins; besides having a single helical TM region, the activity of these proteins usually is associated to their lateral dimerization in cell membranes [37]; thus, bitopic proteins represent the minimum protein unit that crosses biological membranes. In the present study, we performed both experimental and computational analyses of a bitopic TM helical polypeptide, HokC. This peptide is a toxin that kills *Escherichia coli* cells that express it [38], constituting a convenient system to identify critical residues for its toxic function (e.g., loss-of-function mutations will allow cells to growth). The size of this toxin is also convenient to identify single and multiple mutations, since the sequence of the whole gene may be obtained in a single read by any DNA deep sequencing technology available. We provide experimental evidence that HokC accumulates multiple compensatory mutations that are found as correlated mutations in the HokC family multiple sequences alignment. These correlated mutations are twice as much frequently found in transmembrane proteins than in the globular ones, which is accompanied by a lower local density of residue packing in transmembrane proteins compared with globular proteins. Our results together with previous experimental results support the idea that transmembrane proteins are more tolerant to loss-of-function mutations.

## 2. Results

### 2.1. Sensitivity of Experimental Screening

Under growing conditions, *E. coli* cells repress HokC expression to prevent cell death. To disrupt this cellular control, the HokC gene was cloned in the pEXT22 plasmid under the tac promoter; the plasmid also harbors the lacI^Q^ repressor, to ensure maximal repression of the tac promoter. Hence, this expression system guarantees no transcription leakiness of the gene under the tac promoter, which is important to study the effect of this gene expression on cell survival. To derepress the tac promoter from the lacI^Q^ repressor, isopropyl-beta-D-thiogalactoside (IPTG) is commonly used. The chromosomal copy of hokC has 3 ATG codons; we noticed that over-expression of the ORF including the 3 ATG codons did not kill all cells; on the contrary, the *hokC* gene expressed from the second ATG had more toxic effect on *E. coli* cells (data not shown); hence, we used that short version of *hokC* in our experiments. To determine how much IPTG is required to activate the expression of HokC, we used a range of IPTG concentrations (see Methods) and a dilution factor of 0.25 × 10^−2^; hence, if no colonies were detected it meant that the IPTG was preventing the growth of at least 25 times the initial cells exposed to IPTG. We observed than in all, but one, tested IPTG concentrations, *E. coli* cells did not grow (see Appendix A). Since we did not observe any difference in cell viability at different levels of IPTG induction, we assumed that for a mutant to be detected in our system, this has to reproduce the effect of having HokC expression repressed, i.e., we would mainly detect loss-of-function mutations. The mutants that reduced up to 25 times the toxicity of HokC would be detected as wild type.

### 2.2. Mutagenesis of HokC

To reduce the size of the screening, the 23 amino acid residues of the TM region of HokC was mutagenized in regions. For instance, a 3-residue region will generate 30 single point mutations to 1000 multiple mutations (we mutated each position for 10 other residues, see Methods) that will likely be identified by screening 1000 clones or more. Therefore, we selected an average of 1000 isolated colonies for each of the seven mutated regions of the TM region of HokC and classify their phenotypes (see Figure 1). We defined as a wild-type phenotype those cells that upon expression of a HokC mutation no cell colony was observed and, a mutant phenotype corresponds with cells expressing a HokC mutant that upon expression allow the growth of cell colonies (see Methods). The number of colonies analyzed for each of the seven mutated regions and the observed phenotypes are presented in Appendix A. Note that from this first line of results, we may anticipate that regions II (residues 7–9) and VI (residues 19–21) are less likely to contain loss-of-function mutations than the other regions.

After isolating and pooling the DNA from these colonies, we obtained 2,266,368 DNA reads with mutant phenotype and 1,881,708 DNA reads with wild-type phenotype. The sequencing procedure identified mutations beyond the targeted TM region of the protein (all single-residue mutations found in this study are presented in Appendix A). Yet, the occurrences of mutations beyond position 24 (105,301 sequences contained mutations above this position), where the TM region ends, are rare and, consequently, were not taken into account in our analysis (see Appendix A). The incorrect sequence-phenotype assignments were identified following the CHISPAs procedure using a rate of experimental error of 4% (see Methods). Appendix A summarizes all significant single mutants for HokC that rendered a mutant and wild-type phenotype. Two quantitative traits are expected for every position: the number of mutations that rendered a wild-type phenotype (tolerance) and the number of mutations rendering a mutant phenotype (intolerance). We defined as a critical residue any position in the protein sequence for which the ratio of intolerant over tolerant mutations was larger than 1. Our results indicate that none of the residues in the TM region of HokC are critical for its function, yet 19 single-point mutations at 13 different residues eliminate its function (see Appendix A); these are referred to as deleterious or loss-of-function mutations.

We observed that any amino acid substitution (e.g., Ala for Val or Ile for Trp or any other substitution at any given position) in the HokC rendering a mutant phenotype was also found to render a wild type phenotype (see Appendix A). These observations indicate that the position where the substitution takes place is relevant (an Ala for Ile mutation at i-position in the transmembrane region of HokC will not have the same effect if it occurs at j-position) and/or that HokC is able to tolerate many of these mutations. In fact, 4 out of the 13 single-residue substitutions identified to render loss-of-function mutations were found as substitutions in the multiple sequence alignment in the HokC family, suggesting that such natural variants included in the HokC family should have tolerated the mutation if the toxic activity was conserved. We will next explore this idea.

Our experimental design allowed us to identify multiple mutations: HokC variants that include more than one point mutation (see Methods). Among these multiple mutations, we detected compensatory mutations, indicating any combination (double, triple, and so on) of single loss-of-function mutations that showed a wild-type phenotype. Table 1 shows the most frequently observed compensatory mutations in our study; for a full list of these compensatory mutations, see Appendix A. It is noticeable that residue 7 is the only residue in region II that presented deleterious mutations and was the position most frequently observed among compensatory mutations (see Table 1); this result explains the observation about region II (residues 7–9) presenting most of the wild type phenotypes (see Appendix A). All 19 mutations rendering a mutant phenotype (see Table 1) may be compensated (see Appendix A), providing an explanation for the high tolerance of the TM region of HokC to maintain the toxic function of this peptide.

The observed combinations of deleterious single mutations (see Appendix A) that occurred in our experimental set up rendering a wild-type phenotype that were considered compensatory mutations. The table only shows compensatory mutations that are present more than 100 times in our experimental setup. Please note that these compensatory mutations may be present in combination with other tolerated mutations (see Methods); for the list of all compensatory mutations see Appendix A. For a full list of single-point mutations observed in compensatory mutations, see Appendix A.

Interestingly, residue Cys15 tolerated every mutation. Since the previously reported Cysteine to Serine tolerated mutation at that position is conservative and several of the mutations identified at this position were not conservative, we performed a site-directed mutagenesis of this Cys15 residue by three different residues (Cys15Ser, Cys15Glu and Cys15Ala) to validate the tolerance for HokC toxic function of these mutations; our site-directed mutagenesis validated the saturation mutagenesis observations at this position (data not shown).

The orientation of HokC in the TM region is important for its activity. To test for the orientation of the TM region of single (Met7Trp or Ile12Ser) and multiple (Met7Trp-Ile12Ser) mutations of HokC that rendered mutant and wild-type phenotypes, respectively, we fused GFP or phoA to the C-terminus of these mutants. Such constructs have been previously reported to assess the orientation of both N- and C-terminus of TM regions of *E. coli* proteins [39]. As control, we fused GFP or phoA to the wild-type sequence of HokC. Our results showed that the GFP fusions (to wild type or any of the mutants) eliminated the toxic activity of HokC upon induction (see Appendix A). Alternatively, phoA fusions kept the activity of wild type and every mutant tested (see Appendix A). Accordingly, phoA and not GFP fusions, displayed enzymatic activity (see Appendix A). These results indicated that HokC has its C-terminus oriented towards the periplasmic space and that the mutants kept this orientation and the level of expression of the wild type sequence.

In summary, our experimental results revealed that HokC tolerates all single point mutations by accumulating multiple compensatory mutations. This result suggested that: (i) sequence conservation analysis may show low correlation with deleterious mutations, and (ii) TM regions have structural features that allow for accommodating multiple compensatory mutations. To test these hypotheses, we next performed a computational analysis of the HokC protein family and on TM proteins in general.

### 2.3. Are Critical Residues in the TM Region of HokC Conserved?

Using a sequence alignment reported for the HokC family derived from PFAM (see Methods), only one residue (Cys15) identified in the TM region of HokC was invariant (data not shown). To test if this lack of relationship between critical residues and invariant character of residues is the consequence of using an alignment not optimized for TM proteins, we generated a multiple sequence alignment (MSA) with the 148 protein sequences of the PFAM family PF01848 using TM-COFFEE (see Appendix A). Our results indicate that only residue Thr17 was invariant and Val24 presented some degree of conservation, yet none of these positions are critical for protein function. This MSA was also analyzed to compute conservation scores based on the rate4site algorithm (see Methods). According to this analysis (see Appendix A), residues 1, 15, and 17 show the lowest mutability (conservation score ≥8) in the TM region of HokC; furthermore, modifying the parameters of rate4site, it was noted that some correlation between experiments and conservation could be found (data not shown). We explored a third method, PROVEAN (see Methods), which predicted positions 1, 12, and 13 to include deleterious mutations (see Appendix A). Interestingly, position 13 presented substitutions in the MSA that rendered a deleterious effect in our experimental screening (see Table 1). These results confirmed the expected poor correlation between sequence conservation and the loss-of-function mutations in HokC.

One possible mechanism to maintain function without conserving amino acids is by compensatory mutations, i.e., multiple mutations that compensate the deleterious effect of individual mutations. Hence, it is expected that natural variants of HokC may have accumulated compensatory mutations if they were to keep the biological function of HokC. To test this idea, we compared the mutability of each position in the HokC family alignment with that observed in our mutagenesis experiment. As shown in Appendix A, the MSA included residue substitutions at positions K2, V6, A13, I14, V19, A21, and A22 that, in our experimental, data rendered a mutant phenotype (deleterious mutations in Appendix A). This result supports the notion that these loss-of-function mutations must have been compensated if the homologous proteins of HokC should keep their toxic function. To test this idea, we identified all the multiple mutations in the MSA for the HokC family that harbored deleterious mutations for HokC and observed that 91 out of 148 protein sequences included this class of multiple mutations (see Appendix A). Thus, correlated mutations in the HokC family correspond with compensatory mutations identified in our screening. To study whether this is a particular property of HokC or a general trend of TM proteins, we decided to extend our analysis to other TM proteins.

### 2.4. Compensatory Mutations Correlate to High Order Residue Contacts in HokC

According to the expected helical structure of the TM region of HokC, residues that are closer than four residues apart in the sequence may be close in the three-dimensional structure; hence these may be suitable to accommodate compensatory mutations. In agreement with this idea, we observed compensatory mutations in residues that are close at the sequence level (see Table 1). Furthermore, it has been shown that the TM region of HokC may be engaged in the formation of a homodimer as inferred from the mutagenesis of Cys15 for Serine [40]. Our results revealed compensatory mutations between residues far away in the TM region (e.g., positions 6 and 7 with positions 13 and 12, respectively), suggesting that these residues may interact when these are at different monomers; otherwise, an unusual bend on the helix has to be assumed for these residues to interact within the same monomer, which may prevent this region to fully traverse the membrane. The recent prediction reported for the HokC monomer by AlphaFold software version 2, indicates that this TM region does not present an unusual bend in the helix [41], in agreement with the idea that positions 6 and 7 with positions 13 and 12 in HokC monomers participate in the dimerization.

Thus, compensatory mutations in HokC are in agreement with the helical structure of this TM peptide and revealed some other residues that may participate in the dimerization of HokC.

### 2.5. Implications for TM Proteins

Our results indicate that compensatory mutations accumulate among the HokC family of toxins. It has been shown that the loss-of-function single-point mutations may be reverted by combining these with other deleterious mutations [42]. Such mutations are referred to as compensatory mutations that usually correspond with residues close in the 3D structure of proteins [43]. Based on these observations, we wondered whether these mutations accumulated among residues close in the 3D structure of TM proteins (these proteins are structurally classified as mainly alpha or mainly beta) and compared these with globular proteins that presented these same structural classes (see Methods). Our results indicate that TM proteins tend to favor, at least twice as much, the presence of multiple mutations between nearby residues in the 3D structure of proteins (see Figure 2).

To evaluate if the observed increased rate of compensatory mutations is associated with the difference in compactness of TM versus globular proteins, we carried out an analysis of the residue contacts in these two groups of proteins. We observed that as proteins (both globular and TM proteins) change in size, the number of three-dimensional contacts among residues increases proportionally (see Figure 3). This indicates that both globular and TM proteins present a constant packing density, with similar average number of contacts per residue for globular (5.4) and TM proteins (5.4).

In an attempt to identify local differences in packing between these classes of proteins, we looked for maximal cliques in their residue contact maps. Maximal cliques are those cliques (group of residues that are all in contact in the 3D space) that are not part of any larger clique, hence correspond with the densest regions within proteins. We observed that TM proteins accumulated small maximal cliques (size 3) more than globular proteins (see Figure 4). Thus, the set of TM proteins analyzed are less densely packed than the globular proteins as a consequence of reducing the number of large maximal cliques.

Finally, we analyzed the spherical angles between contacting hydrophobic residues (see Methods) to test if this difference in packing may be associated with differences in the arrangement of contacting hydrophobic residues, i.e., we aimed to compare the core of TM proteins with those of globular proteins that belong to the same structural class. To quantify this, we used the Haussdorff distance that estimates the overall difference of two sets of vectors; in this case, hydrophobic residues that are close in distance were represented in vectors, each element in the vector include the angle between the hydrophobic pair of residues. We observed that the orientation of contacting hydrophobic residues of TM proteins and globular proteins differs; particularly, globular proteins (see Figure 5A) tend to have on average smaller Haussdorff distances among their hydrophobic contacting residues compared with TM proteins (see Figure 5B), yet with larger dispersion. Besides the trend presented in Figure 5A,B, we also noticed that 57% of every pair of globular protein analyzed had identical orientation between contacting hydrophobic residues while this occurred in only 18% of the TM proteins. Despite these differences, we observed a group of globular and TM proteins with mainly alpha helical compositions (with the same CATH classification) that showed a very similar contacting geometry (see Figure 5C; for instance structural class 1.10.405.10 or 1.20.5.110). These results indicate that while there is a trend to maintain the geometrical arrangement of hydrophobic residues in globular proteins more than in TM proteins, there are some exceptions to this trend.

## 3. Discussion

Experimental data derived from saturation mutagenesis of proteins indicates that both TM and globular proteins are more tolerant to mutations than expected from phylogenies; however, these previous studies have not addressed the difference in the tolerance to mutations between these two classes of proteins, if any. The relevance of this comparison is that it may help anticipate which of these proteins may adapt more easily to drugs used to control cell fate or to reveal possible reservoirs for new protein sequences and functions, among others. From sequence analysis, it has been observed that TM proteins tend to present a lower degree of sequence conservation than globular proteins [44,45], yet this observation may be the consequence of the method used to align these sequences rather than a property of these proteins. The pioneer work by Bowie’s group showed that the TM regions were as tolerant to mutations as the globular parts of the diacylglycerol kinase from *Escherichia coli*, despite the fact that most critical active-site residues reside in the cytoplasmic domain [46]; yet the coverage of mutations in this experiment was reduced, preventing to fully identify critical residues or compensatory mutations. More recently, it has been shown for the rat neurotensin 1 D03 GPCR that TM regions accepted more diverse mutations than its globular regions [25]; hence, the authors suggested that TM regions are more tolerant to mutations than globular regions. Whether this applies to all TM proteins requires further investigation. To contribute to address this idea, in the present work, we explored the sequence–phenotype space of a TM protein, HokC from *E. coli*. It is relevant to note that the toxic function of HokC depends on its homodimerization and that while we could infer some aspects of this dimerization, our experimental assay cannot discriminate functional defects as a consequence of the monomer or dimer inactivation.

Our experimental results show that 97% (233 out of 240) of all single mutations expected were detected in our screening and only 19 mutations (8%) of these rendered an inactive (non-toxic) HokC peptide (see Appendix A). Hence, the TM region of HokC tolerates most (92%) single-point mutations. For comparison, the C-terminal domain that lays at the periplasmic space of *E. coli* has been proposed to encode for the toxic domain based on two results: (i) the absence of mutations that alter protein function at the N-terminus and (ii) the substitution of the C-terminal region by the phoA resulted in a non-toxic protein [40]. Here, we show that the TM region actually encodes for positions that, upon mutation, alter protein function and that fusing HokC variants to GFP renders an inactive protein. Hence, our results show that HokC toxicity depends on the N-terminal domain and that such domain is more tolerant to mutations than those previously reported for the C-terminus domain. Furthermore, this rate of tolerance for the TM region of HokC is larger than in previous experimental reports showing that globular proteins only tolerate 30–40% of all possible single point mutations [47]. To evaluate whether this is a property of HokC or if this is a general property of TM regions, we performed complementary computational analysis.

In this regard, it has been noted that the core of globular proteins and that of the TM regions are mainly composed of hydrophobic residues, yet different forces drive this similarity in composition. Particularly, globular proteins are subjected to the hydrophobic collapse [48] while the folding of TM regions is commonly assisted laying the hydrophobic residues inside the lipid membrane [49]. This difference suggests that TM regions may tolerate any hydrophobic mutations, yet our results indicate that not every hydrophobic residue is tolerated in the TM region (see Appendix A). This indicates that more complex rules for protein folding take place at the TM region.

This high tolerance to mutations is accompanied with a low degree of sequence conservation observed in the TM region the of HokC family (see Appendix A). Our results indicate that in the case of the HokC family, this sequence diversity is the consequence of combining multiple mutations that harbor deleterious single amino acid mutations (see Table 1, Appendix A). Such multiple mutations may reduce the conservation of many positions in the HokC family and consequently, methods based on sequence-conservation scores fail to properly identify deleterious mutations in this family of toxins. To study the nature of this capacity of TM proteins to accumulate compensatory mutations, we compared the correlated mutations observed between globular and TM proteins and observed that TM proteins tend to accommodate twice as much correlated mutations as globular proteins (see Figure 2). This observation was then compared with the protein packing properties of TM and globular proteins. It has been shown that globular proteins have a constant atomic density [50], i.e., globular proteins with different folds and different sizes all have a similar average number of atoms per volume within a crystal. We have previously reported that the number of contacting residues in the 3D structure of proteins reproduces this phenomenon [51]. Here, we extend these observations to TM proteins and observed that TM proteins have a similar linear trend in the number of contacting residues than globular proteins (see Figure 3). Yet, we observed local differences in the packing of TM and globular proteins, where globular proteins tend to accommodate more residues per unit volume (see Figure 4). This trend is consistent with the observation that TM proteins tend to incorporate voids within their core to fulfill their biological function (e.g., channels [52]) while voids in globular proteins are destabilizing [53] and, consequently, tend to be avoided. Alternatively, voids in any protein have been proposed to locate where proteins are more flexible [54]. From that perspective, our results may be interpreted as TM proteins being more flexible. Thus, our computational analysis shows that TM proteins are locally less densely packed than globular proteins.

In agreement with this concept, we observed that the more dense packing in globular proteins is related to their regular orientation of contacting residues (see Figure 5A,B). In contrast with these observations, geometrical similarities of contacting helix–helix pairs in globular and TM proteins have been reported [55]; here, we show that the density of contacting residues among proteins in the mainly alpha-helical family of proteins are consistent with these previous observations (see Figure 5C). These results indicate that while there are similarities between alpha-helical TM and globular proteins, overall globular proteins tend to vary less the packing in their core than TM proteins. Relevant to these observations is the idea that proteins fold to a minimum energy accessible by densely packing their residues [56]. A solution to this packing problem may be the regular packing proposed by Kepler in the XVII century [57]. Our results provide evidence that globular proteins packed their residues in a more regular way than TM proteins, suggesting that these may approach Kepler’s conjecture. In agreement with these observations, a recent study observed that globular proteins seem to follow Kepler’s arrangement [58]. Thus, these observations indicate that globular proteins tend to maintain a regular packing to comply with the hydrophobic collapse during protein folding. On the contrary, TM proteins allow for more compensatory mutations and have less regular packing than globular proteins; whether this packing affects the mutability of TM proteins deserves further investigation.

Finally, our results complement previous observations about the prevalence of compensatory mutations at sectors in protein structures [59]. Sectors are the regions where compensatory mutations lay in the protein structure that are linked to protein function, with different sectors controlling different biochemical properties of proteins. More recently, it has been noted that in many cases, proteins tend to have a single sector that is dominated by sequence conservation; thus, the relevance of correlated mutations is diminished in those protein regions [60]. Here, we found that the TM region of a toxin that binds to another TM region (homodimerizes) has one sector (TM region accumulates large number of compensatory mutations) with low sequence conservation (see Table 1 and Appendix A). These results suggest that sectors in TM proteins may have different properties than those in globular proteins; this deserves to be further explored.

In summary, we presented a systematic mutagenesis and deep sequencing of the TM region of a bitopic protein, the toxin HokC, to explore its structure–function relationship. We observed that most mutations are tolerated, in agreement with the low degree of sequence conservation of this family of toxins. This poor sequence conservation has an impact on the reliability of prediction methods aimed to identify critical residues. We observed that this family of toxins, and TM proteins in general, tend to accumulate mutations among contacting residues more than globular proteins do. The density of packing between globular and TM proteins may be associated with this trend, by revealing that contacts between residues within membranes follow rules different from those observed in globular proteins. Future mutagenesis of TM proteins may help reveal such rules.

## 4. Materials and Methods

### 4.1. Strains and Reagents

The bacterial strains used in our studies were *Escherichia coli* MC4100 Δ(argF-lac)U169 araD139 rpsL150 relA1 flbB5301 deoC1 ptsF25 rbsR; *E. coli* XL1-Blue supE44 hsdR17 recA1 endA1 gyrA96 thi-1 relA1 lac-; *E. coli* DH5α supE44 ΔlacU169 (φ80 lacZ DM15) hsdR17 recA1 endA1 gyrA96 thi-1 relA1. The alkaline phosphatase activity assay was performed in the CC118 strain and the GFP activity on the BL21(DE3)pLysS strain.

The plasmid pEXT22/frg-hokC containing the gene hokC starting at the second ATG was used as template for both PCR random mutagenesis and for the site-directed mutagenesis. The plasmids for the expression of HokC fused to GFP or phoA were pGFPe and pHA1-yedZ, respectively.

### 4.2. Mutagenesis

Site-directed mutagenesis on the coding region of HokC trans-membrane region was performed using the QuikChange Site-Directed Mutagenesis Kit (Agilent Stratagene, Santa Clara, CA, USA). To that end, we designed a strategy to mutate the TM region of HokC at 7 different groups of neighbor residues as summarized in Appendix A. The following libraries of oligonucleotides were used for this goal:Region 1
R1 Forward 5′ GGA GAA GAG AGC AAT G NNS NNS NNS NNS NNS ATG ATT GTC GCC C 3′
R1 Reverse 5′ GGG CGA CAA TCA T NNS NNS NNS NNS NNS CAT TGC TCT CTT CTC C 3′Region 2
R2 Forward 5′ GCA GCA TAA GGC G NNS NNS NNS GC CCT GAT CGT CAT C 3′
R2 Reverse 5′ GAT GAC GAT CAG GGC SNN SNN SNN CGC CTT ATG CTG C 3′Region 3
R3 Forward 5′ GGC GAT GAT TGT C NNS NNS NNS GTC ATC TGT ATC ACC G 3′
R3 Reverse 5′ CGG TGA TAC AGA TGA C SNN SNN SNN GAC AAT CAT CGC C 3′Region 4
R4 Forward 5′ GTC GCC CTG ATC NNS NNS NNS ATC ACC GCC GTA GTG 3′
R4 Reverse 5′ CAC TAC GGC GGT GAT SNN SNN SNN GAT CAG GGC GAC 3′Region 6
R6 Forward 5′ CTG TAT CAC CGC C NNS NNS NNS GCG CTG GTA ACG 3′
R6 Reverse 5′ CGT TAC CAG CGC SNN SNN SNN GGC GGT GAT ACA G 3′Region 7
R7 Forward 5′ CGC CGT AGT GGC G NNS NNS NNS ACG AGA AAA GAC CTC TG 3′
R7 Reverse 5′ CAG AGG TCT TTT CTC GT SNN SNN SNN CGC CAC TAC GGC G 3′
where S stand for G or C nucleotides and N for any of the four nucleotides. Note that these oligonucleotides will generate mutant codons with SNS composition coding for 10 (L, P, H, Q, R, V, A, D, E, G) out of the 20 conventional amino acid residues. In this way, the number of variants to be screened is reduced and at the same time keeping the diversity of physicochemical properties of the amino acid residues. Please note that each pair of oligonucleotides will hybridize at the corresponding regions that are targeted in the mutagenesis experiment. For instance, the oligonucleotides for region 1 include a 5′ tail (GGA GAA GAG AGC AAT G) required for hybridization that includes the first coding codon (ATG) of the gene followed by 5 codons that are mutated by SNS and followed by a tail in the 3′ end (ATG ATT GTC GCC C) for hybridization purposes. For the site-directed mutagenesis reactions we followed the instructions of the manufacturer: 50 ng of plasmid (pEXT22/frg-hokC), a pair of mutagenic oligonucleotides (125 ng), 1 μL dNTP mix, 5 μL of 10× reaction buffer and 2.5 U of Pfu Turbo DNA Polymerase (Agilent Technologies, Santa Clara, CA, USA) in a 50 μL total volume.

To obtain the HokC mutants Met7Trp, Ile12Ser, and double mutants Met7Trp and Ile12Ser, the QuikChange Lightning site-directed mutagenesis Kit (Agilent Technologies, Santa Clara, CA, USA) was used. The following oligonucleotides were used for this goal:7MxWForw:5′GCAGCATAAGGCGTGGATTGTCGCCCTGATCG 3′
7MxWRev:5′CGATCAGGGCGACAATCCACGCCTTATGCTGC3′
12IxSForw:5′CGATGATTGTCGCCCTGAGCGTCATCTGTATCACC3′
12IxSRev:5′GGTGATACAGATGACGCTCAGGGCGACAATCATCG3′

For GFP fusions, both plasmid pGFPe and PCR products were digested and ligated using XhoI and BamHI restriction enzyme sites. For phoA fusions, the PCR product and plasmid pHA1-yedZ were digested ligated with XhoI and KpnI.

### 4.3. Selection of Clones

To select the hokC variants with wild-type and mutant phenotypes, we performed the following procedure. *E. coli* cells were grown in Luria broth with kanamycin to select for those carrying the plasmid expressing hokC mutations. The plasmid, pEXT22, includes a non-leaky promoter induced by IPTG. The over-expression of hokC was achieved by adding IPTG to the media; this would kill cells expressing a wild-type-like HokC activity. However, cells expressing a mutation critical for HokC activity will grow. All our mutagenesis experiments were performed on a short version of hokC starting from the second ATG codon. To select colonies for sequencing, we looked for isolated colonies; for that end, we used Corning square BioAssay dishes (245 mm × 245 mm of area) (Merck, Kenilworth, NJ, USA).

### 4.4. Sensitivity of Screening

The expression system is reported not to leak transcripts of the genes cloned into the system. To test this and to evaluate how much transcription of the hokc gene was required to kill cells, we conducted a dose–response experiment, where IPTG was added to the media in different concentrations: 0.01, 0.05, 0.1, 0.2, 0.4, and 0.8 mM. *E. coli* DH5a cells were grown overnight to reach a cell density measured at 600 nm of 0.65 measured with a spectrophotomer Genesys 10S UV-Vis (Thermo Scientific, Waltham, MA, USA). These cells were diluted by a factor of 0.25 × 10^−4^ and 100 mL of this dilution were plated on Petri dishes with LB + Kan 10 mg/mL with or without IPTG at different concentrations: 0.01 mM, 0.05 mM, 0.1 mM, 0.2 mM, 0.4 mM, 0.6 mM, and 0.8 mM. These cells were grown for 19 h at 37 °C and the number of colonies that grew in these conditions were counted on a Freedom EVO 150 robotic station using the Pickolo software version 3.5 (SciRobotics, Kfar Saba, Israel.

### 4.5. Sequencing

To sequence mutants in the trans-membrane coding region of hokC, we implemented the following procedure. Colonies with wild-type or mutant phenotypes were picked and grown overnight in 3 mL of LB media with kanamycin 10 mg/mL (Sigma-Aldrich, Estado de Mexico, Mexico). These colonies were pooled in 2 groups according to their origin: cells with a wild-type and mutant phenotypes. From these pools, DNA was extracted. Thus, two pools of plasmids were obtained: from wild-type and mutant phenotype colonies. From these DNA molecules, the mutated hokC region was amplified by PCR to generate the amplicons used for sequencing; the final size of the PCR products was 450 bp. This sample was mixed at equimolar ratios and sequenced at the “Unidad Universitaria de Secuenciación Masiva de DNA-UNAM” using MySeq from Illumina company, with the MySeq reagent kit (Illumina, San Diego, CA, USA) version 2 for 500 cycles, 250 nt each read. Note that the hokC gene is smaller than the reads, thus we will be able to identify the full-length gene sequence of every mutant. TrueSeq DNA PCR-free sample preparation Kit (Illumina, San Diego, CA, USA) was used to add the adapters to our amplicons, without fragmenting the amplicons. Since this sequencer has the capacity to generate 107 DNA reads and the number of bacterial colonies to be sequenced is substantially smaller than this number (103), the experiment could generate thousands of clusters with exactly the same sequence. However, only 80% of the amplicons may have the same sequence and thus, we mixed our amplicons with sequences provided by the “Unidad Universitaria de Secuenciación Masiva de DNA-UNAM”.

### 4.6. Activity of PhoA Fusion Proteins

Strains expressing phoA fusions were grown overnight and inoculated into 50-mL cultures of Luria broth with antibiotic (50 μg/mL ampicillin) at 37 °C to reach an OD at 600 nm of 0.4; then, cells were induced with arabinose (final concentration of 0.2%) and grown for 1 h. The activity assay was carried out as described before [39]. Briefly:Centrifuge 1.2 mL of the bacterial culture in Eppendorf tube.Wash cells in cold WB and resuspend pellet in 1.2 mL cold PM1 buffer.To permeabilize the cells, add 100 μL chloroform and 100 μL 0.05% SDS to 1 mL of the washed cells, vortex for 10 s, and incubate for 5 min at 37 °C. Then place tubes on ice for 5 min. After the chloroform has settled, transfer 100 μL of the upper phase of the bacterial suspension to a 96 plate well.To start the reaction, add 50 μL of the pNPP solution (0.15% in 1 M Tris–HCl, pH 8.0) to the bacterial suspension and incubate at RT until yellow color develops. Add 50 μL 2N NaOH to stop the reaction. Record incubation time and OD at 405 nm for each sample.Calculate enzymatic activity in relative units (A) according to the following formula:
A = 1000 × (OD405sample − OD405control well)/(OD595 sample − OD595control well)/t (min) of incubation

### 4.7. Sequence Data Analysis

DNA reads were trimmed using the Phred algorithm implemented in seqtk (seqtk trimfq option); this process eliminated low quality bases from both ends of the DNA sequences. Then, these fastq files were transformed to fasta files using seqtk (seqtk seq –a option).

The relative frequency of each mutation (*F*(*mut*_i_)) was quantified by the following formula:*F(mut_i_)* = 100 × (*WT_i_* − *MUT_i_*)/(*WT_i_* + *MUT_i_*)(1)
where *WT_i_* corresponds to the number of times the i-mutation (*mut_i_*) was found with a wild-type phenotype and *MUT_i_* is the number of times the i-mutation (*mut_i_*) was found with a mutant phenotype. Then, an ISPA was identified if |*F(mut_i_)*| ≤ Experimental errors. Note that *F(mut_i_)* may be positive or negative, indicating whether the mutant is over-represented in mutant phenotypes or wild-type phenotypes, respectively.

### 4.8. Sequence Alignment

PFAM alignments were obtained from the PFAM web site. By counting the number of sequences that maintain the same residue than the reference sequence (HOKC_ECOLI) the residue conservation score was derived. The same set of sequences was used to align them using TM-COFFEE, an optimized algorithm and substitution matrix for TM proteins [61].

The identification of conserved and critical residues was performed using the Multiple Sequence Alignment generated for the HokC family and the conservation scores were computed based on the rate4site algorithm as implemented in the ConSurf server [62]. Alternatively, PROVEAN was used as an alternative method to identify functionally relevant substitutions [63].

### 4.9. Correlated Mutations Index

Two data sets were used for this analysis: (i) Globular set: 150 globular proteins including different folds and PFAM domain families [64] (see Appendix A) and (ii) TM set: 593 TM proteins from TOPDB [65] (see Appendix A). For each entry in each data set, a multiple sequence alignment (MSA) was obtained from the HSSP database [66]. Additionally, every contacting residue was identified using a 5Å distance criterion as we have previously described elsewhere [67]. Finally, every combined mutation for every contacting residue was identified from the MSA. In this case, each of the 400 possible amino acid pairs were identified and normalized according to number of residue pairs of each kind observed for each protein. For instance, if protein P presented 30 times the pair Ala–Ala and this Ala–Ala pair was mutated 15 times in the MSA, the normalized frequency of correlated mutations for the Ala–Ala pair in protein P is 50% or 0.5. This is the value reported as the correlated mutation index of a protein. The codes to compute this mutation index and datasets are available at [68].

### 4.10. Analysis of Contacts in Proteins

To compare the degree of compactness between globular and TM proteins, we used two larger sets for globular and TM proteins, LG (see Appendix A) and LTM (see Appendix A) sets, respectively. For each set, we computed the size and order of the contact map derived by identifying as contacting residues those closer than 5 Å in at least one pair of atoms as described above. Then, we adjusted the size (number of residues in a given protein) versus the order (number of contacts between residues in a given three-dimensional structure of a protein) to a linear equation using the gnuplot function fit [69]. The difference on the slopes of these two data sets represents the level of difference in packing between these classes of proteins. The size and order for each chain of PDB entries in each dataset and codes are available at [68].

To determine the type of arrangement these proteins adopt upon folded, we compared the spherical angles of clusters of residues. Briefly, every amino acid in a protein and their contacting residues were identified; then, the angles between the central residue and its neighbors were calculated. The angle values obtained for each set were compared using the Haussdorff distance as implemented by Java Topology Suite [70]; to compute the minimum Haussdorff distance for every pair of proteins, we used a simulated annealing algorithm. The codes to compute the spherical coordinates and the minimum Haussdorff distances and associated datasets are available at [68]. Only proteins from the same CATH class with a difference in length no bigger than 20 residues were used for our analysis.

Finally, the number of residue cluster classes (RCCs) of size 3, 4, and 5 were computed as previously described by our group (software version 1 to generate RCCs is available at [71]) and accumulated. Briefly, residue-contacts at 5Å apart were identified and the maximal cliques of size 3, 4, and 5 were quantified.

## Figures and Tables

**Figure 1 ijms-22-10359-f001:**
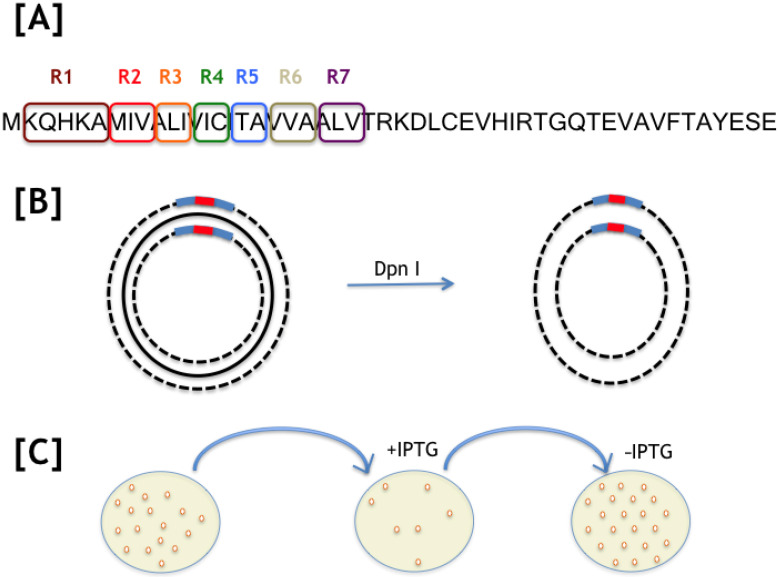
Mutagenesis strategy. (**A**) Seven regions were selected to mutate the ORF coding for HokC; the full sequence of HokC is shown and the regions are marked. (**B**) Oligonucleotides (blue bars) were designed to introduce mutants (red bars) using a QuickChange strategy; the plasmid harboring the wild-type sequence for HokC was amplified (indicated by a punctuated line) and the original plasmid was eliminated by digestion with Dpn I (see Methods for details). (**C**) The plasmids harboring the desired mutations were transferred to competent *E. coli* cells and each colony obtained was replicated into two plates, one with (+IPTG) and another without IPTG (−IPTG), the inducer of HokC expression; cells growing in IPTG harbor a mutation that inactivated the HokC activity and those not growing harbored a mutation that did not affect HokC activity.

**Figure 2 ijms-22-10359-f002:**
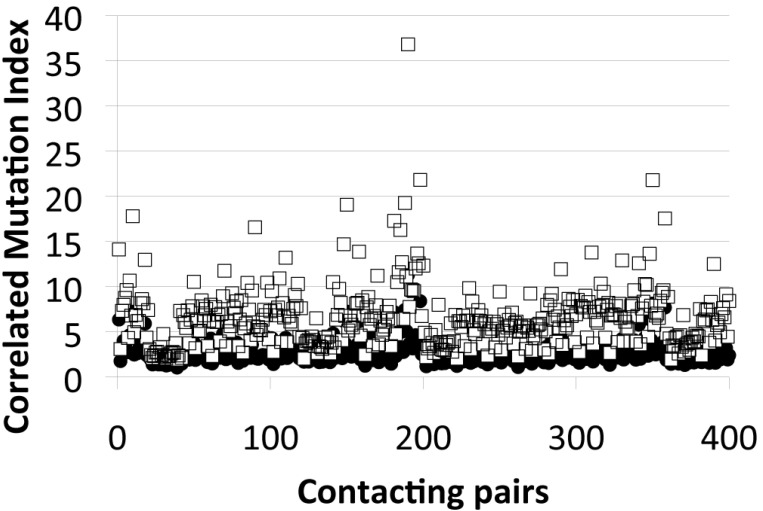
Correlated mutation index of globular and transmembrane proteins. The normalized frequency for all 400 residue pairs at distance of 5 Å in the three-dimensional protein structure (represented in x-axis) that were simultaneously mutated as observed in multiple sequence alignments for their corresponding protein families (correlated mutation index) is presented for both, globular (black circles) and transmembrane (white squares) proteins.

**Figure 3 ijms-22-10359-f003:**
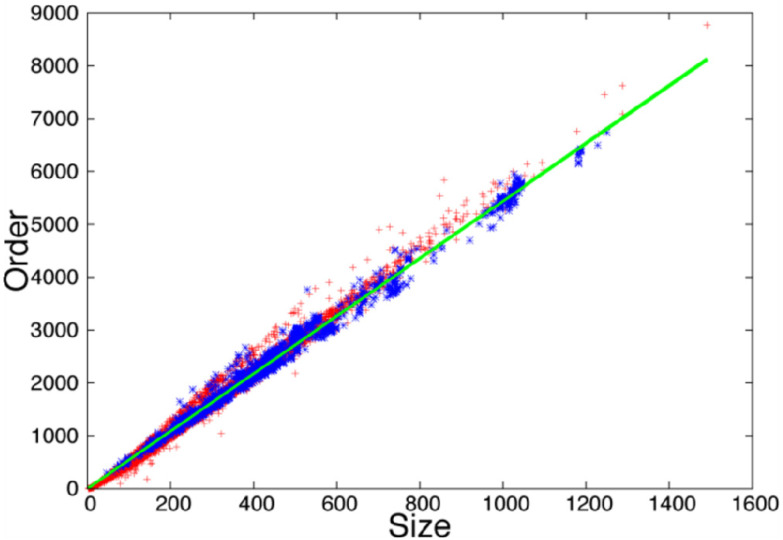
Density of residue contacts for globular and transmembrane proteins. Protein structures were transformed into contact maps at 5 Å to obtain the number of residues (Size) and the total number of reside contacts (Order) for each protein analyzed (see Methods). Size and Order are plotted for both globular (+) and transmembrane (+) proteins. The green line represents the best linear adjustment to both data sets and has a slope of 5.4. The plot was generated using gnuplot.

**Figure 4 ijms-22-10359-f004:**
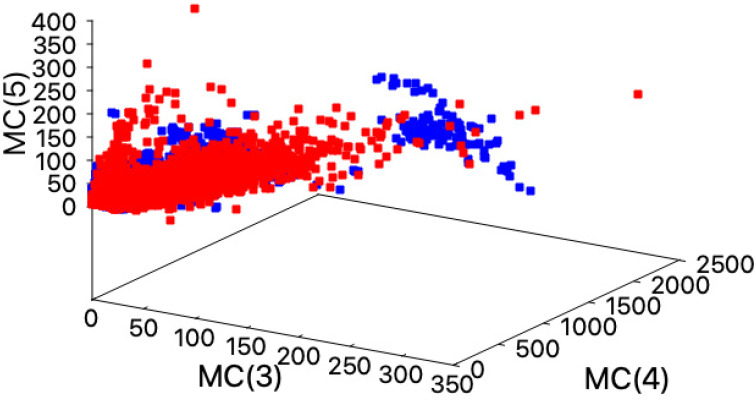
Maximal cliques observed in globular and transmembrane proteins. Protein structures were transformed into contact maps at 5 Å to identify the maximal cliques including 3, 4, or 5 residues using Tomita algorithm (see Methods); maximal cliques correspond with the protein regions where residues are highly packed. Maximal cliques occurrences of size 3, 4, and 5 (axis labeled MC(3), MC(4), and MC(5), respectively), are presented for both globular (◼) and transmembrane (◼) proteins. Please note the cumulus of blue squares on the right side of the image, which include the maximal cliques of size 3 that are accumulated in transmembrane proteins.

**Figure 5 ijms-22-10359-f005:**
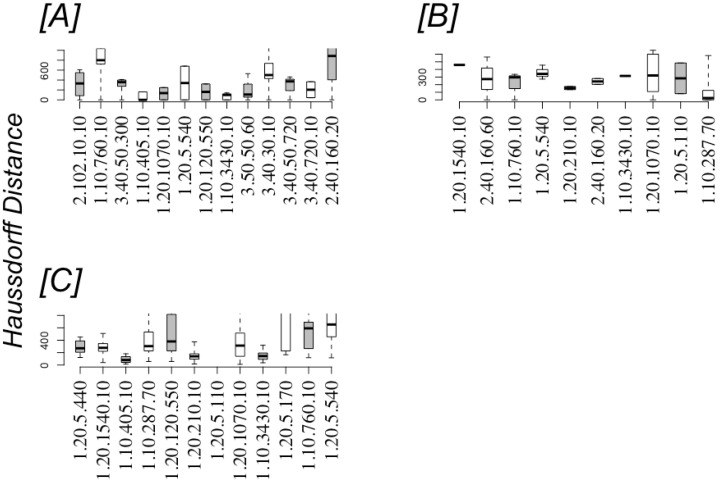
Geometrical differences between globular and transmembrane proteins. The Haussdorff distance (see Methods) was calculated for each protein structure present in the indicated CATH classes on the x-axis for globular (**A**) and transmembrane (**B**). This comparison was conducted also for pairs of globular and transmembrane proteins with the same CATH class with alpha helical structure (**C**). The differences are plotted as boxes, where the median is presented as a horizontal line within the box and the horizontal lines away from the box denote the minimum and maximum values of these distances per CATH class. To facilitate the visualization of these trends, the y-axis value range was ≤700.

**Table 1 ijms-22-10359-t001:** Compensatory mutations in the TM region of HokC.

Combined Mutations (Experimental)	Counts	Combined Mutations (MSA)	Counts
M7W, I12S	613	V13I, A6T	3
I12S, I14S	317	V19L, A6T	8
L11P, I12S	276	A22T, V19L	81
M7W, I12C	221	A6T, K2M	1
M7W, I14S	220	A22S, V19L	2
M7W, L11P	184	A22T, V13I	1
I12S, V19G	145	V19L, V13I	11
I12S, A22T	136	A21T, V19L	5

## Data Availability

All the newly generated data is available as supplemental data in this publication and/or is available as indicated in reference [68]. The previously published data and software used in this study are included as references [41,65,66,69,70,71].

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
