# Peer review of "Saturation Mutagenesis of the Transmembrane Region of HokC in Escherichia coli Reveals Its High Tolerance to Mutations"

_ijms, 2021, doi:10.3390/ijms221910359_

Round 1

Reviewer 1 Report

The authors have adequately addressed my previous concerns.

Author Response

We appreciate the final comments about our work regarding the moderate English changes. Thanks to this note, we have made some further changes to our text to correct some grammar issues we detected.

Reviewer 2 Report

The authors have addressed my concerns. It can now be accepted for publication.

Author Response

We appreciate the final comments about our work regarding the fine/minor English changes required. Thanks to this note, we have made some further changes to our text to correct some style issues we detected.

Reviewer 3 Report

In this work, the authors performed a systematic saturation mutagenesis of the transmembrane region of HokC and observed that 92% are tolerated on single-point mutations. Further, they provided experimental evidence that HokC accumulates multiple compensatory mutations that are found as correlated mutations in the multiple sequence alignment for HokC family. The work is interesting and the following comments could be considered for improvements.

  1. It will be informative to discuss the stability change of point mutations obtained with the methods available in the literature (E.g. MPTherm-Pred; J Mol Biol 2021;433(11):166646)
  2. The selected 10 amino acids for mutations did not have aromatic residues. This could be discussed.

Author Response

In this work, the authors performed a systematic saturation mutagenesis of the transmembrane region of HokC and observed that 92% are tolerated on single-point mutations. Further, they provided experimental evidence that HokC accumulates multiple compensatory mutations that are found as correlated mutations in the multiple sequence alignment for HokC family. The work is interesting and the following comments could be considered for improvements.

1. It will be informative to discuss the stability change of point mutations obtained with the methods available in the literature (E.g. MPTherm-Pred; J Mol Biol 2021;433(11):166646)

We appreciate the suggestion. The MPTherm-Pred method only performs predictions on PDB entries; HokC protein structure is not deposited in the PDB, yet it is available from the AlphaFold2 website: https://alphafold.ebi.ac.uk/entry/P0ACG4. We found an alternative method (http://biosig.unimelb.edu.au/mcsm_membrane) that also predicts the stabilizing/destabilizing effects of single-point mutations on transmembrane proteins and allows the protein structure to be included in a file. Using this method we observed that 76.7% of the single-point mutations rendering a wild-type phenotype had destabilizing effects (http://biosig.unimelb.edu.au/mcsm_membrane/results_stability/1631919191.64), while 75.88% of the single-point mutations rendering a loss-of-function phenotype presented destabilizing predictions (http://biosig.unimelb.edu.au/mcsm_membrane/results_stability/1631919700.55). If the loss-of-function mutations were characterized by destabilizing effects, we would have expected a notable difference between the two sets of mutations, which was not the case. Our dataset does not establish the stability of the mutations and consequently are not useful to improve on methods aimed to predict the stability of single-point mutations. Based on these results, we decided not to include any of these results in our work. 

Additionally, we also run the predictions on the server reported above to identify pathogenic mutations. These mutations represent mutations associated with the development of a disease in humans, and if this concept could be generalized, it should correctly predict any mutation in transmembrane proteins that affect the normal function of the cell or tissue or organism. We learned that all the single-point mutations that rendered a wild-type (http://biosig.unimelb.edu.au/mcsm_membrane/results_pathogenicity/1631932215.85) and mutant (http://biosig.unimelb.edu.au/mcsm_membrane/results_pathogenicity/1631932695.54) phenotype in our experimental setup were predicted as pathogenic. Thus, these results are consistent with the results we mentioned in our work, about the incapacity of current methods aimed to correctly predict critical residues for protein function. As we noted in our work, our experimental results may be used by other groups to improve on their prediction methods. Considering that, we decided not to include these negative results in the current version of our work.

Please note that the links provided will maintain the results for 7 days, starting on September 17, 2021. If the reviewer wants to see these results, we can provide them.

Finally, we did some changes to the text, to correct some grammar/style errors and these changes are marked in red.

This manuscript is a resubmission of an earlier submission. The following is a list of the peer review reports and author responses from that submission.

Round 1

Reviewer 1 Report

Ortiz et al. argues that low conservation in the transmembrane region of proteins is accompanied by compensatory mutations and irregular packing, and these observations assert that the transmembrane protein region can tolerate most mutations. I read the manuscript several times to understand the author's valuable and interesting work, but I didn't get any intuitive insight from it.

In particular, it was very limited to obtain information about the meaning of the primary amino acid sequences listed by the author. Moreover, authors  performed a computational analysis of the HokC protein family and on TM proteins in general to test their hypotheses. However, experimental proof of such computational analysis is essential. Overall, unfortunately, I cannot support the publication of this manuscript in a journal.

On the other hand, I believe that in order to improve the manuscript, it is necessary to clearly indicate the position of amino acids based on the three-dimensional structure of the protein and to explain its functional involvement. Moreover, I believe that the results obtained through computer analysis should be supplemented with experimental evidence.

Author Response

  1. Ortiz et al. argues that low conservation in the transmembrane region of proteins is accompanied by compensatory mutations and irregular packing, and these observations assert that the transmembrane protein region can tolerate most mutations. I read the manuscript several times to understand the author's valuable and interesting work, but I didn't get any intuitive insight from it.

We appreciate the comment from this reviewer. We recognize that the way we presented our work may have caused confusion. We have made several changes to clarify the significance of our work. Among others, we changed the title of our work to reflect that one of the main contributions of our work is the mutagenesis of HokC. Based on these results, we report several computation analyses in an attempt to understand the high tolerance observed in this protein to non-functional mutations. We first show that sequence alignment fail to correctly identify non-functional mutations; we argued that this is the consequence of the alignment metrics. Changing the metrics we observed that some predictors were able to correctly identify some non-functional mutations, yet most were not correctly predicted. We observed that the compensatory mutations identified in our experimental screening were correlated mutations in the multiple sequence alignment. These results suggested a possible explanation about the high tolerability we observed in the TM region of HokC: TM regions would have correlated mutations more often than globular proteins. Since the three-dimensional structure of HokC protein has not been solved, we decided to explore if this trend observed in the TM region of HokC was also observed in other TM proteins. As the reviewer correctly summarized, correlated mutations tend to accumulate among TM proteins, more than in globular proteins. This higher probability of finding correlated mutations, correlates with lower packing density in TM proteins. These results altogether can explain the high tolerability of TM proteins to mutations; having a less dense packing allow them to accommodate more easily mutations without affecting the structure and consequently can maintain function despite mutations. Thus, the original experimental results with HokC, higher tolerability to non-functional mutations than globular proteins, find a computational explanation on correlated mutations in the HokC family and this aspect is also present in many other TM proteins.

  1. In particular, it was very limited to obtain information about the meaning of the primary amino acid sequences listed by the author. 

We agree with this reviewer. Taking the sequence alignment to infer positions that will not tolerate mutations by sequence conservation was not possible, the alignment does not reveal any useful information in that a regard. However, it did show that the correlated mutations in the alignment were also found in our mutagenesis experiment as compensatory mutations. We believe such information is relevant and lead us to pursue this correlated mutations in other TM proteins. We provide the multiple sequence alignment for HokC and its family in Supplemental Figure S7 where the reader can verify our observations.

  1. Moreover, authors  performed a computational analysis of the HokC protein family and on TM proteins in general to test their hypotheses. However, experimental proof of such computational analysis is essential. Overall, unfortunately, I cannot support the publication of this manuscript in a journal.

The aim of our work was to experimentally explore the tolerability of TM region of HokC to loss-of-function mutations. Based on the high-tolerability we observed, we decided to analyze sequence and structure of HoKC family members and TM proteins in general, respectively, in an attempt to understand what were the basis of this tolerability. Our computation results offer a possible explanation, which we argue is consistent with the function as pores or signal transducers of TM proteins. 

  1. On the other hand, I believe that in order to improve the manuscript, it is necessary to clearly indicate the position of amino acids based on the three-dimensional structure of the protein and to explain its functional involvement. Moreover, I believe that the results obtained through computer analysis should be supplemented with experimental evidence.

Unfortunately, HokC protein does not have its three-dimensional structure solved yet. As noted above, the computational analysis was performed to propose an explanation to the experimentally observed high tolerability to lack-of-function mutations in the TM region of HokC. 

Reviewer 2 Report

The authors present an interesting study to understand the tolerance of mutations in the transmembrane region. They report a combined experimental and computational study. The findings are complementary yet an extension of the previous study by Halabi et al. This study focuses on sequencing the TM 423 region of a bitopic protein where there is a single helical TM region. 

The authors should clarify/improve the following points:

  1. Conclusions should specify that tolerance is present upon mutation in the single transmembrane domain. As the nature of tolerance in multiple transmembrane domains was not compared simultaneously.
  2. The functional component suggested by the author needs more experimental study to confirm. The author may consider tone down the suggestion of the functional aspect of this study.
  3. Figures are a little challenging to read particularly, figures 2, 3, 4. The author may consider a significant improvement.
  4. The author should consider moving table 1 to the supplemental Information and summarize the detailed findings in the text.

Author Response

  1. The authors present an interesting study to understand the tolerance of mutations in the transmembrane region. They report a combined experimental and computational study. The findings are complementary yet an extension of the previous study by Halabi et al. This study focuses on sequencing the TM region of a bitopic protein where there is a single helical TM region. 

We appreciate this comment. We want to clarify that while we do refer to the work by Halabi and collaborators, their work did not differentiate globular and transmembrane proteins. Our results argue that globular proteins follow the rules reported by Halabi, but not the TM proteins. Furthermore, the goal of our study was not to extend that work, but to systematically mutagenized HokC to identify loss-of-function mutations and then evaluate the tolerability of HokC TM region. Since Halabi’s claims are relevant to our results, it is reasonable for us to cite their work. By reading the reviewers comments, we realized that the title of our work did not emphasize the experimental aspect and this led to some confusion. Hence, we decided to change the title of our work and make some other changes to help solve this problem.

The authors should clarify/improve the following points:

2. Conclusions should specify that tolerance is present upon mutation in the single transmembrane domain. As the nature of tolerance in multiple transmembrane domains was not compared simultaneously.

We appreciate the note. We have changed the Discussion section to clarify this aspect of our work: “It is relevant to note that the toxic function of HokC depends on its homodimerization and that while we could infer some aspects of this dimerization, our experimental assay cannot discriminate functional defects as a consequence of the monomer or dimer inactivation”.

3. The functional component suggested by the author needs more experimental study to confirm. The author may consider tone down the suggestion of the functional aspect of this study.

We appreciate the note. We have changed the Abstract and Introduction sections to eliminate the functional speculation we included in the first version of our work.

4. Figures are a little challenging to read particularly, figures 2, 3, 4. The author may consider a significant improvement.

We appreciate this note. In this new version of our work, we have changed the figure legends to facilitate the reading of these images.

5. The author should consider moving table 1 to the supplemental Information and summarize the detailed findings in the text.

We have moved Table 1 to Supplementary material and now it is referred to as Table S4. The rest of other tables were renumbered to accommodate this new supplementary table.

Round 2

Reviewer 1 Report

Thanks for the author's sincere reply, some of my concerns have been resolved. I certainly recognize that the author's research is worthwhile. However, I think it is very limited to understand the relationship between the structure and function of the HokC family only with the amino acid sequence information mentioned in the manuscript. Like the author's opinion, I am also sorry that the structure of HokC is currently not available. It is expected that the author's assertion will be more reliable when it is provided along with the 3D structure in the future. It is judged that there is insufficient information for the current manuscript to be published on IJMS. Accordingly, I regret that I cannot support the publication of the manuscript.